# Effects of Low-Lipid Diets on Growth, Haematology, Histology and Immune Responses of Parr-Stage Atlantic Salmon (*Salmo salar*)

**DOI:** 10.3390/ani14111581

**Published:** 2024-05-27

**Authors:** Byoungyoon Lee, Junoh Lee, Saeyeon Lim, Minjae Seong, Hanbin Yun, Sijun Han, Kang-Woong Kim, Seunghan Lee, Seong-Mok Jeong, Mun Chang Park, Woo Seok Hong, Se Ryun Kwon, Youngjin Park

**Affiliations:** 1Department of Aquatic Life Medical Sciences, Sunmoon University, Asan 31460, Republic of Korea; yoon980912@naver.com (B.L.); znaks67@naver.com (J.L.); kathy0709@naver.com (S.L.); tjdalswo221@naver.com (M.S.); kenzan743@daum.net (H.Y.); air0987@naver.com (S.H.); srkwon@sunmoon.ac.kr (S.R.K.); 2Aquafeed Research Center, National Institute of Fisheries Science (NIFS), Pohang 37517, Republic of Korea; kangwoongkim@korea.kr (K.-W.K.); lsh@kunsan.ac.kr (S.L.); smjeong1@korea.kr (S.-M.J.); 3Gangwon State Inland Water Resource Center, Chuncheon 24210, Republic of Korea; parkmc800@korea.kr (M.C.P.); hws90@korea.kr (W.S.H.)

**Keywords:** Atlantic salmon, lipid, parr stage, growth, RNA-seq

## Abstract

**Simple Summary:**

This study investigated the effects of three artificial diets used in aquaculture, two for rainbow trout and one for Atlantic salmon, on the growth rate, organ responses, gene expression and immune responses of parr juvenile Atlantic salmon. The diets differed with respect to their carbohydrate, protein and lipid levels, with the lipid levels being the main focus of this study. We found no significant differences in the growth, feed efficiency, immunity, histology or gene expression of Atlantic salmon parr between the rainbow trout and salmon feed groups. Therefore, rainbow trout feed, which has a relatively low lipid content, could be used as a rearing feed for juvenile parr-stage Atlantic salmon, bringing potential economic benefits.

**Abstract:**

Lipids in fish diets provide energy and play important roles in immunity and metabolism. Atlantic salmon, a species that migrates from freshwater to seawater, requires high energy, especially during smoltification. Juvenile teleosts have low lipid requirements, and a high dietary lipid content is known to have negative effects on their growth and digestion. Therefore, this study evaluated the effect of two commercial rainbow trout feeds (low-lipid, 13.41% and 14.6%) on the growth and immune responses of early parr-stage Atlantic salmon compared to commercial salmon feed (high-lipid, 29.52%). Atlantic salmon parr (weight: 14.56 ± 2.1 g; length: 11.23 ± 0.44 cm) were randomly divided into three groups and fed either one of two commercial rainbow trout feeds (RTF1 and RTF2) or the commercial salmon feed (ASF) for 12 weeks. At the end of the feeding trial, growth, haematology, histology and gene expression analyses were performed. There were no significant differences in weight gain rates or feed efficiency between the groups (*p* > 0.05). Superoxidate dismutase, glutathione peroxidase, lysozyme and immunoglobulin M activities were not different among the experimental groups (*p* > 0.05). A histological examination of the liver and intestinal tissues showed no pathological symptoms of inflammatory response or lipid accumulation in any of the groups. In an intestinal transcriptome analysis using RNA-seq, the expression levels of several genes linked to lipids, immune-related proteins, cytokines and chemokines did not differ significantly between the groups (*p* > 0.05). Commercial rainbow trout feed with low lipid content has no clear negative impact on the development of Atlantic salmon during the early parr stage (14.5 to 39.6 g). This study provides basic information for the development of economical feed for early parr-stage Atlantic salmon.

## 1. Introduction

Atlantic salmon (*Salmo salar*) is one of the most valuable farmed fish species worldwide [1,2]. Currently, the annual production of Atlantic salmon exceeds 2.7 million tonnes, with a steady increase since 2000. Atlantic salmon aquaculture accounts for more than 32% of marine and coastal fish farming globally [3], highlighting its significant commercial importance. Atlantic salmon has been actively researched because of its economic value. The demand for Atlantic salmon in Korea is growing, but its supply relies almost entirely on imports [4]. Thus, several salmon farming complexes are being developed in Korea. To reduce the cost of salmon farming, significant efforts have been made to develop salmon feed using domestic ingredients.

Atlantic salmon is a carnivorous species classified as an anadromous fish [5]. They inhabit freshwater during the initial parr stage and then move to seawater through a transformation from parr to smolt (smoltification) [6]. Smoltification induces significant morphological, physiological and behavioural changes that affect their metabolic rate, oxygen consumption and osmoregulation [1,7]. Maintaining an optimal nutritional status is crucial to ensure stable smoltification, which requires increased energy accumulation and efficient lipid utilisation [8,9].

Among the dietary nutrients, lipids have a relatively high energy content and are widely used in high-energy diets [10]. Also, dietary lipids are an important source of essential fatty acids required for the growth, health, reproduction and physical function of fish, including Atlantic salmon [11,12]. A high-fat diet can be applied with farmed salmon because of their excellent ability to utilise dietary lipids as an energy source, especially during smoltification [13,14]. Along with the development of feed with a high energy density, the lipid content of salmon feed has increased [15]. However, the price of fish oil, used as a lipid source, is continuously increasing as the demand for aquaculture feed increases, and it is expected to continue increasing in the future [16,17].

Although high-lipid diets have been shown to be beneficial for the smoltification of Atlantic salmon [18], it is possible that dietary lipids have no impact on the growth or digestive health of salmon, especially at the fry and parr stages [19,20,21]. A study on salmon demonstrated that fish fed high-lipid diets showed no significant differences in growth compared with those fed a low-lipid diet [22,23]. In addition, another study showed that reducing the dietary lipid level of salmon from 24% to 14% at the parr stage did not affect their growth or feed intake [18]. In the case of other carnivorous fish species, including walleye pollock (*Gadus chalcogrammus)*, longsnout catfish (*Leiocassis longirostris Günther*) and white seabass (*Atractoscion nobilis*), feeding on a high-lipid diet during the fry stage had negative effects on their growth, feed intake and digestion [19,20,21]. These findings underscore the importance of tailoring feed formulations to the species and developmental stages.

Rainbow trout, *Oncorhynchus mykiss*, also belong to the Salmonidae and are extensively farmed domestically in Korea. Unlike Atlantic salmon, rainbow trout exclusively inhabit freshwater environments and do not undergo smoltification. Currently, commercial rainbow trout feeds tend to have a lower lipid content than commercial Atlantic salmon feeds. Lipid contents in salmon diets increase when the size of fish closes to smoltification (from 20 to 30.5%) [24,25,26]. But those in trout diets range from 14 to 18% [27,28,29,30]. The use of feed with a low dietary lipid content may achieve economic benefits in salmon feed development. Therefore, the purpose of this study was to determine the feasibility and effects of using low-lipid commercial rainbow trout feed during the parr stage of Atlantic salmon compared with high-lipid commercial salmon feed. 

## 2. Materials and Methods

### 2.1. Experimental Feeds

For the 12-week feeding trial, three commercial feeds were used: two domestic rainbow trout commercial feeds (RTF1, Woosung, Daejeon, Republic of Korea; RTF2, Purina, Seongnam, Republic of Korea) and one imported commercial Atlantic salmon feed (ASF, Aller Aqua, Christiansfeld, Denmark). The proximate compositions of the experimental feeds are listed in Table 1. RTF1 and RTF2 consist of 47.83% and 50.87% protein and 13.41% and 14.60% lipid, respectively. ASF consists of 48.55% protein and 29.52% lipid. The carbohydrate contents of RTF1, RTF2 and ASF were 15.47%, 11.12% and 3.69%, respectively. The feeds were stored refrigerated (4 °C) until use.

### 2.2. Experimental Fish and Management

The Atlantic salmon eggs used in this study were purchased from Benchmark Genetics (Hafnafjordur, Iceland). The eggs hatched into alevins at the Gangwon-do Inland Water Resource Center, and the salmon were raised in five 2500 L tanks. Once grown to parr size, we randomly distributed 900 fish (initial weight: 14.56 ± 2.1 g; initial length: 11.23 ± 0.44 cm) equally into three 2000 L circular tanks with a diameter of approximately 2 m (300 fish/tank). The fish were reared in a recirculating aquatic system with filtration equipment. Prior to the start of the feeding experiment, all fish were acclimated to the experimental conditions by rearing them in the experimental tanks for a period of one week. Three experimental feeds were fed by hand to the fish twice a day until apparent satiation (8:00 and 15:30). The water quality was measured once daily. The water temperature, pH and dissolved oxygen were maintained at 15 ± 0.1 °C, 6.81 ± 0.01 and 7.73 ± 0.06 mg/L, respectively.

### 2.3. Chemical Analyses

The proximate compositions of the feed and fish were analysed according to the Association of Official Analytical Chemists (AOAC) methods. Moisture, ash and lipid contents were determined according to the methods described by [31]. The protein content was analysed using an automatic protein analyser (Kjeltec System 2300, Foss Tecator AB, Hoganas, Sweden).

Fatty acids were extracted using the method described by Metcalfe and Schmitz [32]. The separated fatty acids were analysed using a gas chromatographer equipped with a capillary column (112-88A7, 100 m × 0.25 mm, film thickness 0.20 μm, Agilent Technologies, Santa Clara, CA, USA). Hydrogen was used as the carrier gas, and the oven temperature was increased from 140 to 240 °C at a rate of 4 °C/min. Both the injection and detection temperatures were set to 240 °C. A standard sample of the PUFA 37 component, FAME Mix (Supelco Inc., Bellefonte, PA, USA), was used.

For amino acid analysis, homogenised samples (2–3 mg of protein from the three commercial feeds) were taken and hydrolysed in 30 mL of 6N HCl at 130 °C for 24 h. When the hydrolysis was complete, the solution was diluted to 100 mL with distilled water and filtered through a 0.4 μm soluble syringe filter. The hydrolysed samples were diluted 1:1 and analysed using HPLC.

### 2.4. Sampling

After the 12-week feeding experiment, 30 fish per experimental group were randomly selected and anaesthetised using ethyl 3-aminobenzoate methanesulfonate solution (100 mg/L, Sigma, St. Louis, MO, USA). Their total length and weight were measured to calculate their final body weight (FBW), weight gain rate (WG, %), feed efficiency (FE, %), specific growth rate (SGR, %), condition factor (CF), hepatosomatic index (HSI, %) and visceral somatic index (VSI, %). The survival rate (SR, %) was determined by counting the total number of fish. 

Among all the sampled fish, fifteen fish were taken and stored at −20 °C until the analysis of their whole-body proximate composition. Blood samples were collected from the caudal vein using a syringe. For plasma separation, the collected blood was transferred to a 1.5 mL tube coated with heparin (BD Microtainer, Franklin Lakes, NJ, USA). For histological analysis, the liver and intestinal tissues were extracted and fixed in a 10% formalin solution. The fixed tissues were stored at room temperature without direct sunlight until further analysis. For RNA-sequencing, intestinal tissue was transferred to 1.5 mL microtubes and stored in a deep freezer (−80 °C) until analysis.

### 2.5. Haematological Analysis

The collected blood was centrifuged (4 °C, 1000× *g*, 15 min) for plasma separation. Then, the supernatant (plasma) was collected and stored at −80 °C until analysis. Using ELISA kits (CUSABIO, Houston, TX, USA), superoxidate dismutase (SOD), glutathione peroxidase (GPx), lysozyme (LZM) and immunoglobulin M (IgM) assays were performed according to the manufacturer’s protocol. The optical density of each assay was read at 450 nm using a microplate reader (TECAN Infinite 200 Pro M, Männedorf, Switzerland).

### 2.6. Histological Studies

The intestine and liver tissues were fixed in a 10% formalin solution for a week and then immersed in 95% and 99.9% ethanol for 1 h each. The samples were then transferred to xylene, which was used twice for 1 h each and once for 75 min. The samples were then embedded in paraffin wax. After the paraffin block was produced, it was sectioned to a thickness of 4 μm and attached to a slide. To remove paraffin from the slides, xylene was applied for 2 min each. Afterwards, xylene was removed using 99.9% ethanol for 2 min each, as well as 95% and 70% ethanol for 30 s each. Before staining, the sections were washed for 30 s and stained with haematoxylin staining reagent for 5 min. After they were washed with water for 30 s, the sections were washed in 0.3% HCl alcohol for 3 s and then stained with eosin Y for 1 min and 30 s. Dehydration was performed using 70%, 95% and 99.9% ethanol for 30 s. Transparency was assessed using xylene for 6 min, after which samples were mounted on Canada balsam. Slices of intestinal and liver tissues were prepared and placed on a microscope slide. The tissues were photographed at 200× magnification using THUNDER Imager 3D Tissue (Leica Microsystems, Wetzlar, Germany).

### 2.7. RNA-Sequencing Analysis

From the collected intestinal samples of the RTF1 and ASF groups, the total RNA was isolated using a QIAzol Lysis Reagent and RNeasy Mini Kit (Qiagen, Hilden, Germany) and treated with DNase. RNA was purified using the TruSeq Stranded Total RNA with Ribo-Zero (Illumina, San Diego, CA, USA). The purified RNA was randomly fragmented. The RNA fragments were converted into cDNA via reverse transcription. The cDNA was ligated and amplified via PCR. An insert size of 200–400 bp was selected using a size selection process. Samples were sequenced via paired-end sequencing (101 bp) using a novaseq6000 (Illumina).

Quality control analysis of the raw reads was performed. Artefacts of low quality, such as adapter sequences, contaminant DNA and PCR duplicates, were excluded. Aligned reads were generated after mapping to the reference genome using the HISAT2 program. Transcript assembly was performed using the StringTie software (version 2.2.0). The expression profile was extracted from the expression levels obtained via transcript quantification using FPKM (fragments per kilobase of transcript per million mapped reads) values. RNA-seq data were analysed according to the methods described in previous studies [33,34]. Expressed genes linked to lipid metabolism, immune-related enzymes, cytokines and chemokines were selected to compare the expression levels between the RTF1 and ASF groups.

### 2.8. Statistical Analysis

Growth performance, haematological parameters, the expression of selected genes and intestinal morphology data were expressed as means ± SDs. R Studio (version 2023.06.1524) was used for the statistical analysis and data visualisation. The Shapiro–Wilk test and Bartlett’s test were performed to test the normality and homogeneity of the experimental data, followed by a one-way ANOVA test. The significance (*p* < 0.05) among the experimental group means was compared using Tukey’s HSD test. For non-parametric tests, the Kruskal–Wallis test and Dunn’s test were performed to determine significant differences between the groups.

## 3. Results

### 3.1. Composition of Experimental Feeds and Fish

To understand the compositions of the commercial feeds, the fatty acids (Table 2) and amino acids (Table 3) of experimental feeds were analysed. The contents of ∑SFA in the feed were 7.04% and 7.9% in the RTF1 and RTF2 groups, respectively, and 16.52% for the ASF group. The contents of MUFA in the feed were 2.6% and 2.9% in the RTF1 and RTF2 groups, respectively, and 5.7% in the ASF group. Three commercial feeds had similar amino acid profiles.

Regarding whole-body proximate compositions at 12 weeks (Table 4), there were no differences in the protein, ash and moisture contents among all the experimental groups. However, the lipid content in the ASF group tended to be higher than that in the other groups.

### 3.2. Growth Performance

At the end of the feeding trial, no pathological symptoms were observed in any experimental groups (each group had over a 90% survival rate). The results of the bulk test for the growth of the salmon fed three experimental feeds are shown in Table 5. At 12 weeks, the RTF1 groups had the highest WGR in the bulk test. The results of the growth performance at 12 weeks are shown in Table 6. There were no significant differences in WGR, SGR, FE or CF among all the experimental groups (*p* > 0.05). However, the FI in the RTF1 and RTF2 groups tends to be higher than that in the ASF group. While the HSI in the RTF1 and RTF2 groups were significantly higher than those in the ASF group (*p* < 0.05), there were no significant differences in VSI among all the experimental groups (*p* > 0.05).

### 3.3. Haematological Parameters

The results of the activities of immune-related proteins in the plasma of Atlantic salmon fed the experimental diets for 12 weeks are shown in Figure 1. In SOD, GPx, LZM and IgM, there were no significant differences among all experimental groups (*p* > 0.05).

### 3.4. Histological Analysis

The histology of liver and distal intestine tissues from three experimental groups is shown in Figure 2 and Figure 3, respectively. No pathological symptoms, including lipid accumulation, were observed in the hepatocytes of any of the experimental groups. Table 7 shows a comparison of intestinal morphology parameters among the experimental groups. Based on Figure 3, the morphological parameters among the experimental groups were compared. The villi length, villi thickness, lamina propria width, muscularis thickness and goblet cell counts were similar among all the experimental groups (*p* > 0.05).

### 3.5. Selected *Gene Expressions* in RTF1 and ASF Groups

Table 8 shows a summary of the RNA-sequencing for the RTF1- and ASF-fed groups. Of around 485 million trimmed reads from six samples (three from RTF1 and three from ASF), over 449 million reads were mapped to the Atlantic salmon genome. The average mapping percentage among the samples was 92.54%.

The expression of genes linked to lipids, immune-related enzymes, cytokines and chemokines was profiled and compared between the RTF1 and ASF groups (Figure 4). The gene expression levels in the RTF1 and ASF groups were calculated based on FPKM values. There was no significant difference in the expression levels of genes linked to lipids (*apoa1*, *apoa2*, *fadsd5*, *fadsd6*, *cpt1ab* and *elovl2*), immune-related enzymes (*sod1*, *sod1*, *sod2*, *gpx2*, *gpx1a*, *gpx4a*, *lyz*, *lyg*, *lyg*, *cbr1* and *alox5a*), cytokines (*irf3*, *irf7*, *ifngr1a*, *ifngr2b*, *il2ra*, *il18*, *il22ra2*, *litaf*, *traf6* and *traf2*) or chemokines (*ccl19a.1*, *cxl10*, *ccl19*, *cxcl12a*, *ccl25b*, *ccl20a.3*, *cxcl9*, *ackr3b*, *ackr4b* and *ccr9*) between the RTF1 and ASF groups (*p* > 0.05).

## 4. Discussion

In this study, we aimed to investigate the effects of low-lipid diets on growth, feed efficiency, immune enzymes and gene expression in early parr-stage Atlantic salmon. This study attempted to provide basic knowledge on the development of more economical juvenile parr-stage salmon feed.

In the proximate composition analysis of the one Atlantic salmon and two rainbow trout commercial feeds used in this study, ASF contained higher lipid levels than RTF1 and RTF2, and the protein level was similar (47–50%) in all three feeds. The ash, calcium and phosphorus contents were all similar, but the carbohydrate content was lower in ASF. Fatty acid levels were similar to those of total lipids, with ASF showing higher fatty acid levels than RTF1 and RTF2. Amino acid levels were similar in all feeds. Regarding the whole-body proximate composition, there were no differences in whole-body protein, ash or moisture levels between the experimental groups. However, the ASF group, which had a high dietary lipid content, tended to have higher whole-body lipid levels than the RTF1 and RTF2 groups, showing that there was a close relationship between the proximate composition of the feed and the levels in the whole fish body. These results are consistent with those of lipid studies on brown trout (*Salmo trutta*) and gibel carp (*Carassius auratus gibelio*) [35,36].

Weight gain and the specific growth rate are important growth indicators often used in breeding experiments. The WG and SGR results in this study were not significantly different between the experimental groups. As a result, the growth rate did not increase with the increasing lipid content of the feed, suggesting a limited effect on fish growth. Generally, the feed intake is closely related to the growth rate and feed efficiency [37]. Berrill, Porter and Bromage [22] and Storebakken and Austreng [38] found that the growth rate was significantly affected due to the feed intake. In this study, FI was slightly higher in the RTF1 and RTF2 groups than in the ASF group, but there was no significant difference in FER. The reason for this disparity is likely related to the high-lipid feed and total dietary energy [39]. Studies on juvenile grass carp (*Ctenopharyngodon idella*) [40] and catfish (*Pseudobagrus ussuriensis*) [41] showed no significant differences in the condition factor according to the lipid content of the feed, which is consistent with our study. Based on these results, it appears that a low-lipid feed at the parr stage does not negatively affect growth.

The hepatosomatic index was expressed as a percentage of the total liver weight of the fish [42,43]. And the viscerosomatic index was expressed as a percentage of the total viscerosomatic weight of the fish [44]. In addition, the hepatosomatic and viscerosomatic indices are closely related to lipid accumulation and can be used as health indicators [45,46]. In general, differences in dietary lipid levels are associated with lipid accumulation in the viscera [47]. According to Nanton et al. [48], the HSI increases with increasing dietary lipid levels. However, in this study, the viscerosomatic index did not differ significantly between the experimental groups, although the RTF1 and RTF2 groups showed a higher HSI than the ASF group, which is believed to have been due to the carbohydrate content of the feed. Some studies showed that a high-carbohydrate-diet group had a higher HSI than groups with low-carbohydrate diets, suggesting that high-carbohydrate diets can induce glycogen storage in the liver and subsequently increase liver weight [49,50]. Similar to the present study, a low-lipid diet group showed a higher HSI in the fry and juvenile stages of rainbow trout, grass carp, iridescent shark (*Pangasius hypophthalmus*) and seabass (*Dicentrarchus labrax*) [41,51]. Additionally, Peres and Oliva-Teles [52] and Wang et al. [53] suggested that the HSI increased as the carbohydrate content in the feed increased.

Reactive oxygen species (ROS) generated in the body are highly reactive and can cause oxidative damage to cells [54]. SOD and GPx are antioxidant enzymes that play important roles in the neutralisation and processing of ROS [55]. Therefore, antioxidant enzymes can be used as indicators of biological health [56]. Excessively high dietary lipids can negatively affect the immune response of fish due to tissue lipid peroxidation [57]. In this study, there was no significant difference in plasma SOD levels among the experimental groups. Similar to this study, dietary lipid content did not affect SOD activity in juvenile croaker (*Nibea coibor*) and loach (*Misgurnus anguillicaudatus*) [58,59]. There were no significant differences among the experimental groups in the GPx results, similar to the SOD results. Similarly, different lipid levels in the diet of seabass did not affect the GPx activity in the plasma [60]. Lysozymes are responsible for innate immunity, and they are antibacterial enzymes that protect against microorganisms [61]. In the LZM analysis, no significant differences were observed between the experimental groups. Several studies have shown that dietary lipids above the optimal level did not have a significant effect on lysozyme levels and that lysozyme activity increased as dietary lipids increased from 2% to 10% in juvenile rohu (*Labeo rohita*). However, lysozyme activity was reduced in the group fed 12% dietary lipids [62,63]. Immunoglobulin M (IgM) is a glycoprotein that recognises and destroys disease-causing antigens. In fish, IgM has been found to coat bacteria with a type of mucus [64]. The concentration of IgM can vary, depending on various factors, such as water quality, stress and the size of the fish species [64]. In this study, the IgM levels were the same in all groups. Based on these results, we believe commercial low-lipid rainbow trout feed does not affect the activity of immune-related proteins.

Severe lipid accumulation in the liver is an indicator of a lipid metabolism disorder [65]. Additionally, hepatocytes are very important nutritionally and pathologically, and they can be used as an indicator of nutritional status [66]. In this study, the pathological symptoms of lipid accumulation and inflammation in hepatocytes were not observed in any of the experimental groups. There were no significant differences between groups in terms of liver cell density or cell nuclear location. The intestinal mucosal epithelium is both nutritionally and immunologically important [67]. Excessive lipid accumulation in fish intestines has the potential to destroy cells and damage the intestinal tissues [68]. The typical symptom of the intestinal inflammatory response in fish is that the intestinal villi become shorter, and the thickness of the lamina propria increases, owing to the influx of many immune cells [67,69]. In this study, no inflammatory response to lipid accumulation was observed, and there were no significant differences between any of the groups in the length or thickness of the villi, the thickness of the intestinal muscle layer or the number of goblet cells. The difference in HSI in our study was not thought to be due to lipid accumulation or an inflammatory response, but it may have been influenced by dietary carbohydrates. Therefore, a low-lipid diet does not appear to negatively impact the health of the liver or intestinal tissues.

The intestine has been studied extensively as a central organ for nutrient absorption and pathogen recognition [70]. RNA-sequencing is a technology that can be used to quantify the type and expression levels of RNA in biological samples. RNA-sequencing (RNA-seq) is widely used in fish research [71]. Using RNA-seq, the expression levels of various genes related to lipids, enzymes and immunity were quantified and compared between RTF1 and ASF intestinal tissue samples.

Among the lipid-related genes, several genes with high expression levels (*apoa1*, *apoa2*, *fadsd5*, *fadsd6*, *cpt1ab* and *elovl2*) showed no significant differences between the RTF1 and ASF groups. In Atlantic salmon, *apoa1* and *apoa2* were identified as genes involved in lipoprotein formation, *fadsd5* and *fadsd6* as genes involved in fatty acid transport and *cpt1ab* as a gene that decomposes fatty acids [72]. In another Atlantic salmon study, *elovl2* was suggested to be important for the transcriptional regulation of adipogenic genes [73]. Although there was a difference in the dietary lipid content of the feed, this was not thought to affect the expression of lipid-related genes.

The expression levels of immune-related protein genes (*sod1l*, *sod1*, *sod2*, *gpx2*, *gpx1a*, *gpx4a*, *lyz*, *lyg*, *lygl*, *cbr1* and *alox5a*) were confirmed. There were no significant differences in their expression levels, including SOD, GPx and LZM-related genes, between the RTF1 and ASF groups. This is similar to the results from the plasma analysis, showing that the low-lipid feed did not change the intestinal immune enzymes.

Cytokines, including interleukins, interferons, tumour necrosis factor (TNF) and chemokines, play important roles in the immune system [74]. In the present study, there were no significant differences between the experimental groups in the expression levels of several genes (*irf3*, *irf7*, *ifngr1a*, *ifngr2b*, *il2ra*, *il18*, *il22ra2*, *litaf*, *traf6* and *traf2*). A study on flounder [75] demonstrated that the expression levels of inflammatory cytokine genes, including *il22ra2*, *tnfaip2b*, *irf7* and *irf3*, were higher in the viral haemorrhagic septicaemia virus-infected group (VHSV) than in the control group. Another study on salmon showed the up-regulation of the interleukin 2 subunit beta gene *il2ra* upon stress stimulation (chasing, netting, noise and temperature shock) [76]. Chemokines are known to play a role in regulating inflammation, and they contribute to the immune response. Chemokines regulate the movement of white blood cells and are important mediators of inflammation during the immune response [77]. Kim, Park, Kwon and Park [75] showed that the expression of chemokines and chemokine-related genes (*ccl19a.1*, *cxl10* and *ccl19*) increased in the kidneys of flounder infected with VHSV. In addition, the expression levels of several pro-inflammatory chemokine genes (*ccl19* and *ccl20*) in Atlantic salmon were increased via immunostimulants (often pathogen-associated molecular patterns), and the chemokine genes (*ackr3*, *ackr4* and *ccr9*) were upregulated after infectious salmon’s anaemia virus infection [78]. In the present study, the expression levels of *ccl19a.1*, *cxl10*, *ccl19*, *cxcl12a*, *ccl25b*, *ccl20a.3*, *cxcl9*, *ackr3b*, *ackr4b* and *ccr9* did not differ between the RTF1 and ASF groups. These results suggest that a low-lipid diet does not induce intestinal inflammatory responses or chronic stress.

## 5. Conclusions

In this study, no significant differences were found in the growth, feed efficiency, immunity, histology or gene expression of Atlantic salmon parr between rainbow trout and salmon feed groups. Therefore, it appears that rainbow trout feed, which has a relatively low lipid content, can be replaced as a rearing feed for juvenile parr-stage Atlantic salmon ranging from about 14.5 to 39.6 g. Additionally, the use of feed with a low dietary lipid content may achieve economic benefits. However, further detailed studies using formulated diets are required to determine the specific effects of low-lipid diets before and after the smolt stage.

## Figures and Tables

**Figure 1 animals-14-01581-f001:**
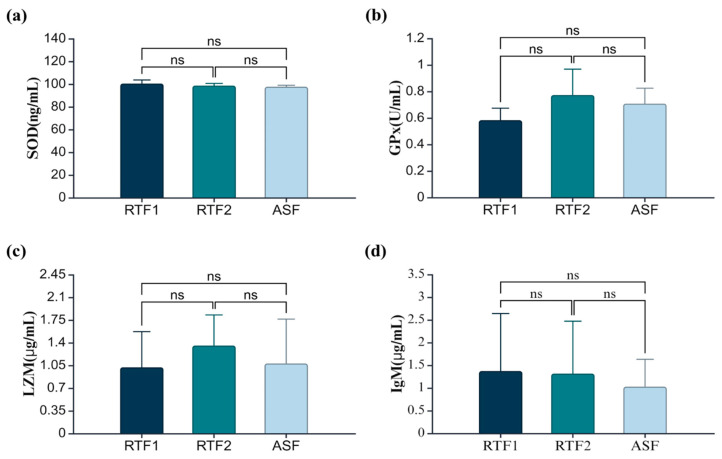
Analysis of immune-related proteins in plasma. The values in the figure are means ± SDs (*n* = 6). ns indicates no significant differences among the experimental groups (*p* > 0.05). (**a**) Superoxidate dismutase (ng/mL); (**b**) glutathione peroxidase (U/mL); (**c**) lysozyme (µg/mL); (**d**) immunoglobulin M (µg/mL); RTF1, rainbow trout feed 1; RTF2, rainbow trout feed 2; ASF, Atlantic salmon feed.

**Figure 2 animals-14-01581-f002:**
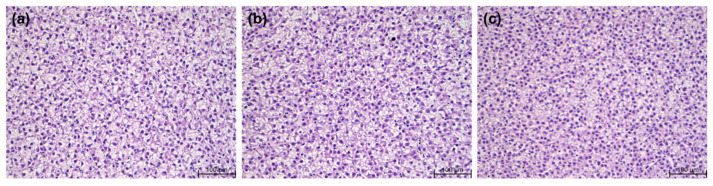
Hematoxylin and eosin stain of the livers of Atlantic salmon fed three experimental feeds for 12 weeks. All images were captured at 200× magnification. Scale bar = 100 µm; (**a**) rainbow trout feed 1; (**b**) rainbow trout feed 2; (**c**) Atlantic salmon feed.

**Figure 3 animals-14-01581-f003:**
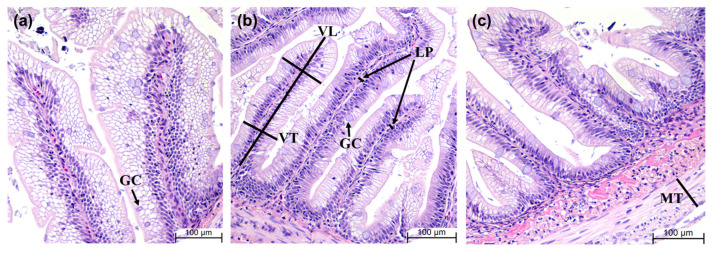
Representative images of intestinal histology (hematoxylin and eosin stain) of experimental fish fed three experimental feeds. All images were captured at 200× magnification. Scale bar = 100 µm; VL, villi length; VT, villi thickness; LP, lamina propria; MT, muscularis thickness; GC, goblet cells. (**a**) Rainbow trout feed 1; (**b**) rainbow trout feed 2; (**c**) Atlantic salmon feed.

**Figure 4 animals-14-01581-f004:**
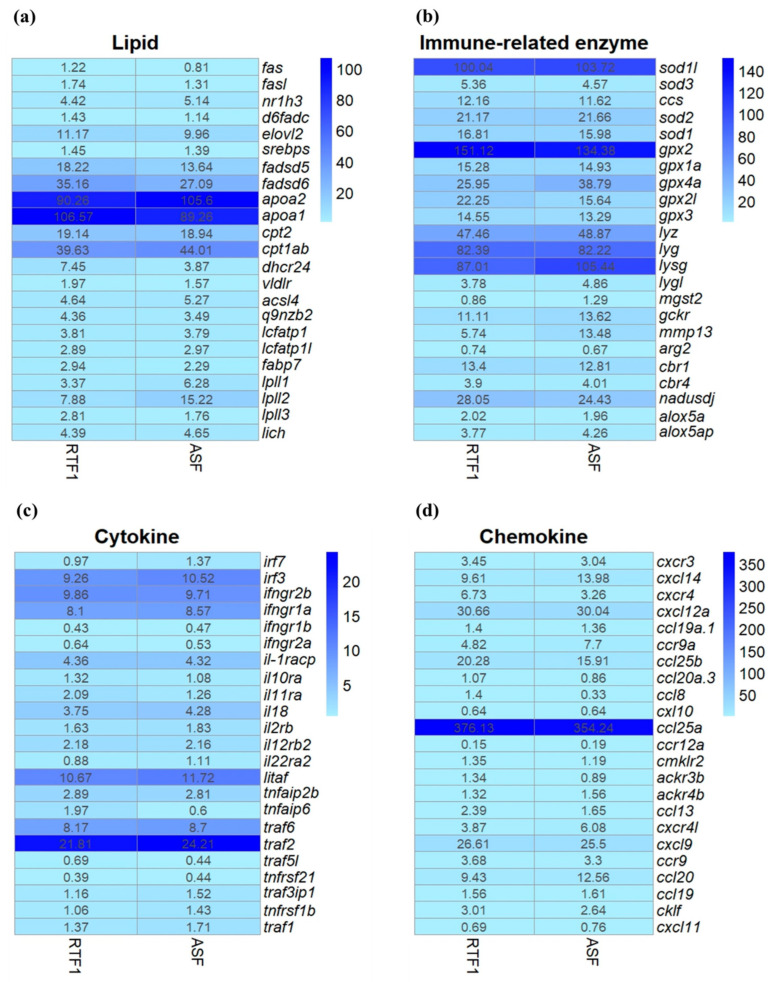
Heatmap of the expression of selected genes from Atlantic salmon intestines. The FPKM values shown in the heatmaps are the averages of three replicates. The colour gradient, from light blue to dark blue, in the heatmap indicates increasing expression levels. (**a**) Lipid-related genes; (**b**) immune-related enzyme genes; (**c**) cytokine-related genes; (**d**) chemokine-related genes.

**Table 1 animals-14-01581-t001:** Proximate compositions of three experimental feeds (%, dry matter basis) ^1^.

	RTF1	RTF2	ASF
Protein	47.83	50.87	48.55
Lipid	13.41	14.60	29.52
Carbohydrate	19.53	16.71	5.74
Ash	11.43	12.96	8.57
Calcium	2.41	3.74	1.02
Phosphorus	1.65	1.85	1.03
Moisture	7.8	4.86	7.62

^1^ Feed samples were analysed in duplicate. Values in the table are means. RTF1, rainbow trout feed 1; RTF2, rainbow trout feed 2; ASF, Atlantic salmon feed.

**Table 2 animals-14-01581-t002:** Fatty acid compositions of three experimental feeds (%, dry matter basis) ^1^.

	RTF1	RTF2	ASF
∑SFA ^2^	7.04	7.90	16.52
∑MUFA ^3^	2.60	2.90	5.70
C18:2n-6 ^4^	0.62	0.59	1.16
C18:3n-6 ^5^	0.15	0.05	0.10
C20:4n-6 ^6^	0.19	0.20	0.39
C18:3n-3 ^7^	0.03	0.04	0.08
C20:5n-3 ^8^	1.37	1.42	2.59
C22:6n-3 ^9^	1.10	1.10	1.77

^1^ The feed samples were analysed in duplicate. The values in the table are means. ^2^ ∑SFA, total saturated fatty acids; ^3^ ∑MUFA, total monounsaturated fatty acids; ^4^ C18:2n-6, linoleic acid; ^5^ C18:3n-6, gamma-Linolenic acid; ^6^ C20:4n-6, arachidonic acid; ^7^ C18:3n-3, alpha-linolenic acid; ^8^ C20:5n-3, eicosapentaenoic acid; ^9^ C22:6n-3, docosahexaenoic acid.

**Table 3 animals-14-01581-t003:** Amino acid compositions of three experimental feeds (%, dry matter basis) ^1^.

	RTF1	RTF2	ASF
Asp ^2^	4.04	3.75	3.86
Thr ^3^	1.66	1.78	1.69
Ser ^4^	1.82	2.20	1.68
Glu ^5^	6.24	6.80	5.91
Pro ^6^	1.67	2.67	1.18
Gly ^7^	2.47	3.40	2.37
Ala ^8^	2.44	2.67	2.35
Cys ^9^	0.29	0.31	0.34
Val ^10^	2.14	2.29	2.18
Met ^11^	0.79	0.78	0.99
Ile ^12^	1.67	1.81	1.76
Leu ^13^	3.15	3.02	3.09
Tyr ^14^	1.34	1.10	1.25
Phe ^15^	1.73	1.64	1.57
His ^16^	1.15	0.96	1.21
Lys ^17^	3.05	3.48	3.15
Arg ^18^	2.42	2.64	2.29

^1^ The feed samples were analysed in duplicate. The values in the table are means. ^2^ Asp, aspartic acid; ^3^ Thr, threonine; ^4^ Ser, serine; ^5^ Glu, glutamic acid; ^6^ Pro, proline; ^7^ Gly, *glycine*; ^8^ Ala, alanine; ^9^ Cys, cysteine; ^10^ Val, valine; ^11^ Met, methionine; ^12^ Ile, *isoleucine*; ^13^ Leu, leucine; ^14^ Tyr, tyrosine; ^15^ Phe, phenylalanine; ^16^ His, histidine; ^17^ Lys, lysine; ^18^ Arg, arginine.

**Table 4 animals-14-01581-t004:** Whole-body proximate compositions of Atlantic salmon fed three experimental feeds for 12 weeks (%, wet matter basis) ^1^.

	RTF1	RTF2	ASF
Protein	16.7	16.8	16.6
Lipid	8.2	9.5	12.1
Ash	2.26	2.47	2.37
Moisture	71.90	69.91	68.23

^1^ A pool of whole-body samples from fifteen fish was analysed in duplicate. The values in the table are means.

**Table 5 animals-14-01581-t005:** Bulk test for growth of Atlantic salmon fed three experimental feeds for 12 weeks.

	RTF1	RTF2	ASF
0-week			
Total number	300	300	300
Total weight ^1^	4000.29	4205.87	4875.95
Mean weight ^2^	13.33	14.02	16.25
12-week			
Total number	283	292	299
Total weight	11,160	11,480	12,720
Mean weight	39.43	39.32	42.54
WGR ^3^	195.79	180.45	161.78

^1^ The total weight (g) is the sum of the weight of all fish in each experimental group. ^2^ Mean weight (g) = total weight/total survival number. ^3^ Weight gain rate (%) = 100 × (final mean body weight − initial mean body weight)/initial mean body weight.

**Table 6 animals-14-01581-t006:** Growth performance of Atlantic salmon fed three experimental feeds for 12 weeks ^1^.

	RTF1	RTF2	ASF
WGR ^2^	188.8 ± 81.9	180.5 ± 62.3	153.7 ± 54.9
SGR ^3^	1.2 ± 0.4	1.2 ± 0.2	1.1 ± 0.3
FI ^4^	7465.9	7515.7	6847.9
FE ^5^	95.4 ± 41.4	98.3 ± 34	112.9 ± 33.4
CF ^6^	1.01 ± 0.07	0.98 ± 0.05	1.0 ± 0.04
SR ^7^	94.3	97.3	99.7
HIS ^8^	1.64 ± 0.34 ^a^	1.58 ± 0.19 ^a^	1.15 ± 0.22 ^b^
VSI ^9^	10.06 ± 2.48	10.88 ± 1.82	11.56 ± 2.66

^1^ The values in the table are means ± SDs (*n* = 30). ^2^ Weight gain rate (%) = 100 × (final mean body weight—initial mean body weight)/initial mean body weight. ^3^ Specific growth rate (%/day) = [100 × (ln final weight − ln initial weight)/days], ^4^ FI (g) = feed intake. ^5^ Feed efficiency (%) = 100 × (wet weight gain/dry feed intake). ^6^ Condition factor = 100 × (fish weight/total body length^3^). ^7^ Survival rate (%) = 100 × (numbers final alive fish/numbers of initial fish). ^8^ Hepatosomatic index (%) = 100 × (liver weight/whole body weight). ^9^ Viscerosomatic index (%) = 100 × (viscera weight/whole body weight). Values with different superscript letters in the same column are significantly different (*p* < 0.05).

**Table 7 animals-14-01581-t007:** Intestinal morphology of Atlantic salmon fed three experimental feeds for 12 weeks ^1^.

	RTF1	RTF2	ASF
VL ^2^	584.7 ± 70.2	520.3 ± 92.2	517 ± 94.3
VT ^3^	186.3 ± 50.3	131.5 ± 29.3	144.2 ± 70.2
LPW ^4^	10.3 ± 1.5	9.3 ± 4.5	5.5 ± 0.7
MT ^5^	81.5 ± 17.0	77.4 ± 30.2	82.4 ± 19.4
GC ^6^	16.7 ± 3.2	14.3 ± 4.0	16 ± 8.9

^1^ The values in the table are means ± SDs (*n* = 8). ^2^ VL, villi length (µm); ^3^ VT, villi thickness (µm); ^4^ LPW, lamina propria width (µm); ^5^ MT, muscularis thickness (µm); ^6^ GC, goblet cells (counts in a villus).

**Table 8 animals-14-01581-t008:** Data of raw, trimmed and mapped reads from RTF1 and ASF groups.

Tissue	Groups	Sample Codes	Raw Reads	Trimmed Reads	Mapped Reads	Mapped %
Intestine	RTF1	RTF1-1	68,405,114	66,770,388	61,208,210	91.67
Intestine	RTF1	RTF1-2	71,528,998	69,981,766	65,108,647	93.04
Intestine	RTF1	RTF1-3	89,345,438	87,279,420	80,649,931	92.4
Intestine	ASF	ASF-1	89,295,688	87,104,434	80,598,127	92.53
Intestine	ASF	ASF-2	89,050,832	86,934,950	80,643,286	92.76
Intestine	ASF	ASF-3	89,541,694	87,765,022	81,503,436	92.87

## Data Availability

The study’s data are contained within the article.

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
