# Peer review of "Effects of Low-Lipid Diets on Growth, Haematology, Histology and Immune Responses of Parr-Stage Atlantic Salmon (Salmo salar)"

_animals, 2024, doi:10.3390/ani14111581_

Round 1
Reviewer 1 Report
Comments and Suggestions for Authors
It is true that the economy of fish farming is very important and feed includes more than 50-60% of current production costs, but the performance and quality of fry production in aquaculture should not be sacrificed for low prices. Of course, the most expensive part of aquatic feed is always protein rather than fat. The manuscript is well-written and easy to understand and read. All in all, please follow the comments/corrections/questions:
· Salmon like other carnivorous fish species needs to include relatively a high lipid content in its diet to supply the body energy (as the authors described in L68-69). Therefore, why the authors chose this topic for this species at this stage of life (parr)? What is scientific justifications?
· L31: I agree but if it is more than the body requirement and the fish in this study need more energy for the smoltification. Although in salmons a large amount of fat initially accumulates in the body cavity and may even increase the final weight and weight gain, it will reduce the yield percentage of the fillet.
· What was the lipid contents in the trial diets? The abstract should be written in such a way that the readers can understand everything at a glance without going into the article.
· Why the feeding duration is high (12 weeks), as 8 weeks is enough and common in aquaculture nutrition studies.
· It is strange for me that most of the studied parameters did not significantly different in high and low lipid diets, since fish at parr stage need a high content of lipid in their feed formulation based on many references. Low fat diets can therefore favorably alter the energy balance equation and it can therefore negatively alter the energy balance equation and decrease body skeletal muscle mass. What is the justification? Maybe since the authors used commercial diets, there are other ingredients that can act as energy for the fish.
· L51: I reckon, this species is replaced with GMO salmon (AquAdvantage salmon) in the world. I checked the reference No.1 and the topic was not mentioned the importance Salmo salar production in the world and it is about metabolisim response in S. salar.
· According to NRC or other references, what is the requirement for salmon parr in terms of lipid and gross energy?
· How much dietary lipid inclusion can pose the negative impacts in salmon?
· L82-83: please give numerical differences
· Another major drawbacks is the experimental diets. What is the justification of the authors to ensure the commercial feed is free from any functional additives that can interfere with the results? Because the formulation of commercial aquafeed is secret. Why the experimental diets did not prepared by the authors, which is commonly performed in aquaculture nutrition.
· No complete dose study was performed in this study (13-14% and 29.5% lipid content) that makes the readers maybe lower or higher lipid content can pose different and even better results in S. salar. Maybe a polynomial test could help to find the optimal lipid levels when there are more lipid content treatments.
· L136: What was the concentration?
· Why the bulk test was not done for growth indices and the authors sampled some limited fish (n=30). What is the scientific reason to sample 30 fish from 300 fish?
· use ×g (RCF) instead of rpm for the centrifugation methods. Plz check it throughout the manuscript.
L302-322: It likes the content to be suitable for introduction rather than discussion.
Author Response
Dear reviewer,
I hope you have a nice day.
Thank you so much for your kind review.
We have revised manuscript based on your comments (please find the yellow highlights in attachment).
Kind regards

Reviewer 2 Report
Comments and Suggestions for Authors
This manuscript (ID 3004735) entitled "Effects of low lipid diets on growth, haematology, histology and immune responses of parr-stage Atlantic salmon (Salmo salar)" investigated the detailed effects of two commercial rainbow trout feed (low-lipid level) and one commercial salmon feed (high-lipid level) on growth performances, hematobiochemical parameters, visceral morphology, and metabolism/immune-related transcriptomic profile in Atlantic salmon during the parr-stage. Both two rainbow trout feed showed no negative influence on juvenile parr-stage Atlantic salmon because of no significant differences in the relevant parameters between rainbow trout and salmon feed groups. Results from this study could highlight the feasibility of low-fat rainbow trout feed as an effective alternative to high-fat salmon feed in Atlantic salmon and provide the necessary foundation for developing effective, low-cost, and sustainable low-fat feed in Atlantic salmon and other salmonid aquaculture species.
The main content of this manuscript is valuable for farmed Atlantic salmon, especially at the parr stage. However, the main content of discussion section contains several redundant descriptions and should be streamlined and reorganized. Also, the authors need to proofread throughout this paper for language to improve readability.
Major comments:
1.Regarding Table 1 (Line 99-100), information on the complete composition of three experimental feeds for Atlantic salmon is missing. The authors should give the relevant information to Table 1 in "2.1 Experimental Feeds" section.
2. In the section of "2.7. RNA-Sequencing Analysis", intestinal samples in the RTF1 group and ASF group were isolated for RNA-seq. Why did not choose hepatic samples in these two groups? Please explain it.
3. Regarding Fig.4, was the representative sections of intestine from ASF group or other groups? Why not show the histological images of intestinal sections from fish fed with three experimental feeds?
4.Insufficient information about RNA-seq data was provided in this manuscript. What about the results of DEGs and the corresponding enrichment analysis (GO and KEGG analysis)? The author should provide more data and information on RNA-seq expression data in the revised paper.
5. The main text of discussion part contains the repeating statements. Please streamline and reorganize the section of "Discussion" with emphasis for better clarifying your findings in this study. For example, Line 328-322 and Line 328-331 contained redundant descriptions or repeated statement. The author could remove superfluous text and concentrate on the main finding and description in the revised paper.
Additionally, the relevant references are missing in some part of this section. For example, in Line 306-308, the corresponding references should be cited in "Previous studies".
6.The authors should check the references format carefully after reading the Instructions for Authors, especially the first letter of co-author name in the references should be capitalized, note the abbreviations of all co-authors, DOI, journal volume information. For example, in reference 1, DOI information should be revised. There were similar errors in the other references. In reference 25, the name of co-authors in the references should be presented as "Du, Z.Y.; Liu, Y.J.; Tian, L.X.; Wang, J.T.; Wang, Y.; Liang, G.Y".
Also, most cited literature in the "Reference" section is rather old, only 15 references published in 2019-2024. For example, reference 14 is published in 1961. Please make sure 50% of the references are within 5 years (2019-2024).
Minor comments:
1. Please specify initial weight of Atlantic salmon in the "Abstract" section (Line 32-33).
2.In Line 205-206, no information on significant differences in the contents of SFA and MUFA content among three experimental diets were included in this manuscript. The authors should check and provide the relevant description in the main text and the legend of Table 2.
3.Results of whole-body nutritional composition in Atlantic salmon
were expressed as mean. According to the statement of "2.8. Statistical Analysis" section, data in this study were presented as mean ± SD (Line 195). Why did not use "mean± SD" in Table 4?
4.Some tables lack the description of "a" and "b". For example, Table 4 and Table 5. More information on the different lowercase letters in the corresponding tables should be provided in the revised legends of tables.
5.Check the symbols for volume unit and mass unit in this study according to the information to related guides. For example, in Line 253-254, please replace "ng/ml" and "ug" with "ng/mL" and "μg", respectively. There were similar errors in Fig.1 and other part of this manuscript. Please check and modify accordingly.
Other errors were presented in the PDF file.
Therefore, this manuscript will be reconsidered after major revision.

This paper (ID 3004735) entitled "Effects of low lipid diets on growth, haematology, histology and immune responses of parr-stage Atlantic salmon (Salmo salar)" is valuable for aquafeed and aquaculture, particularly Atlantic salmon. However, several descriptions in the main text of "Discussion" were redundant. Also there were still several mistakes, such as spelling errors, the presenting symbols for volume unit. It is recommended that the text should be proofread by a native speaker.
Author Response

(The authors gave the same response as above.)

Reviewer 3 Report
Comments and Suggestions for Authors
The work investigated the effects of three commercial diets ( two rainbow trout feeds containing low lipid contents and one Atlantic salmon feed containing high lipid content) on the growth performance, haematology, liver and intestine histology, immune response and gene expression of Atlantic salmon in par size. The results showed that no significant differences in the growth, feed efficiency, immunity, histology or gene expression of Atlantic salmon parr fed with three diets were detected which indicated that the two rainbow trout feeds containing low lipid contents could be used as a rearing feed for Atlantic salmon parr. The work offered a possible solution for the reliance on domestic aquafeeds for salmon culture in Korea. Overall, the experimental design was reasonable, the presentation of results was fine and the discussion was also persuasive. But i just wondered why the low lipid content diets led to higher HSI of salmon parrs than that of the fish fed higher lipid content feed. Generally, carnivorous fish had very limited ability to convert the excessive carbohydrate in the diet to lipids due to their poor compacities on the carbohydrate utilization. The author should provide some specific proofs to support their explanation. In conlusion, i suggested that the manuscript is appropriate for the acceptance of publishment in the Journal after the authors give the reasonable explantation for why the low lipid content diets led to higher HSI of salmon parrs .
Author Response

(The authors gave the same response as above.)

Round 2
Reviewer 1 Report
Comments and Suggestions for Authors
The authors have provided a detail revision and good luck.
Author Response
Dear reviewer,
Thank you so much for your valuable comments.
From your kind advice, this paper becomes very better.
I hope you have a wonderful day.
Best wishes
Reviewer 2 Report
Comments and Suggestions for Authors
This modified paper (Animals-3004735) titled "Effects of low lipid diets on growth, haematology, histology and immune responses of parr-stage Atlantic salmon (Salmo salar)" has been carefully revised in response to the comments of the reviewers. Also, the authors have addressed the reasons for the unchanged section in the list of responses. Therefore, it is recommended to accept this revised version for publication.
Author Response

(The authors gave the same response as above.)
